

# Repertoire of morphable proteins in an organism

Keisuke Izumi, Eitaro Saho, Ayuka Kutomi, Fumiaki Tomoike and
Tetsuji Okada

Department of Life Science, Gakushuin University, Tokyo, Japan

## ABSTRACT

All living organisms have evolved to contain a set of proteins with variable physical and chemical properties. Efforts in the field of structural biology have contributed to uncovering the shape and the variability of each component. However, quantification of the variability has been performed mostly by multiple pair-wise comparisons. A set of experimental coordinates for a given protein can be used to define the "morphness/unmorphness". To understand the evolved repertoire in an organism, here we show the results of global analysis of more than a thousand *Escherichia coli* proteins, by the recently introduced method, distance scoring analysis (DSA). By collecting a new index "UnMorphness Factor" (UMF), proposed in this study and determined from DSA for each of the proteins, the lowest and the highest boundaries of the experimentally observable structural variation are comprehensibly defined. The distribution plot of UMFs obtained for *E. coli* represents the first view of a substantial fraction of non-redundant proteome set of an organism, demonstrating how rigid and flexible components are balanced. The present analysis extends to evaluate the growing data from single particle cryo-electron microscopy, providing valuable information on effective interpretation to structural changes of proteins and the supramolecular complexes.

## INTRODUCTION

Self-replicating (living) species are defined by the presence of a genome that is used to produce a set of proteins and RNAs. The diversity of living entities, from unicellular bacteria to mammals, primarily corresponds to the varying repertoire of proteins. A proteome set limits the morphology and functionality of an organism. Therefore, understanding the atomic details of the folded state and the variability of the fold for each protein in an organism is a long-term goal in structural biology.

We have previously shown that comparison of different proteins and protein families with respect to their structural variability can be comprehensively performed by distance scoring analysis (DSA), in which all intramolecular $C_\alpha$–$C_\alpha$ pair distance variations (score = average/stdev) are considered (*Anzai et al., 2018*) (Fig. 1). DSA is advantageous in terms of reproducibility because no procedures/assumptions associated with superimposition are involved. Also, the amount of numeral usage is far more than the pair-wise based analysis. For example, if we calculate an average root-mean-square

Corresponding author
Tetsuji Okada,
tetsuji.okada@gakushuin.ac.jp

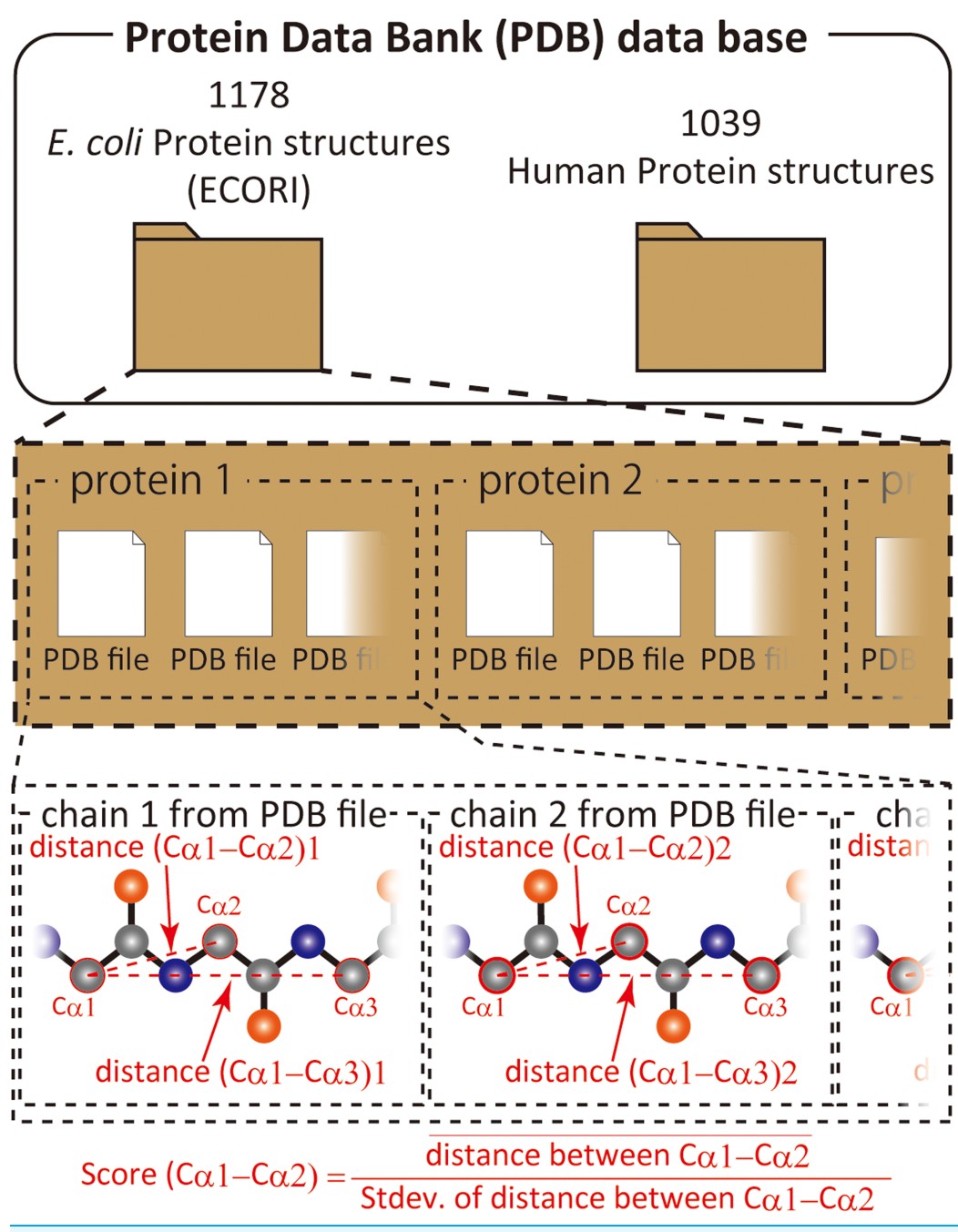

**Figure 1 Scheme of score calculation.**

deviation (RMSD) of $C_\alpha$ positions for 10 structures (chains) of 100 amino acids, the number of used deviation values is 100 positions × 45 structure pairs = 4,500; whereas, DSA uses (100 × 99/2) $C_\alpha$–$C_\alpha$ pair distances × 10 structures = 49,500 distance values. Furthermore, inclusion of new data is simpler in DSA as no pairing is required. The average score for a protein of 100 amino acids is thus obtained as an average of 4,950 scores from 49,500 distances in the case of 10 chain usage. Here we introduce "UnMorphness Factor" (UMF), which is defined as a converged average score for a protein, is an

alternative to conventional RMSD-based comparison. In most cases, the average score of a protein decreases as the number of used chains increases, reaching a nearly steady level, which is assigned as UMF of a protein. Thus, when we successively calculate the average scores using 3, 4, 5 … 9 chains as average score 3, 4, 5 … 9, the last average score for nine chains (average score 9) is defined as UMF (Fig. S1). In the following study, UMFs calculated from X-ray crystallography are denoted as xUMF and those from single-particle cryo-electron microscopy (cryo-EM) are denoted as eUMF.

*Escherichia coli* proteins are the most abundant prokaryotic content in the Protein Data Bank (PDB) (*Burley et al., 2017*). We suppose that species-based construction of a "morphability" map is most achievable, covering a substantial portion of proteins that construct the *E. coli* proteome. The present work is intended to assign a UMF to a protein having a UniProt ID as well as to highlight how experimental coordinates can be used to uncover the repertoire of morphable proteins in *E. coli*. Here we demonstrate that a substantial fraction (~23%) of the *E. coli* proteome can be described as exhibiting "morphness/unmorphness" distribution with well-defined lower/upper limits, and a major cluster of data points. The result is compared to a similar amount (~1,000) of randomly selected human proteins, a part of which was the subject of our previous study (*Anzai et al., 2018*), providing unprecedented insight into the morphness repertoire for a set of natural polypeptides. These analyses of X-ray structures are also used to evaluate the recently accumulating cryo-EM structures.

## METHODS

### Protein lists

ECOLI and HUMAN reference proteome set obtained from UniProt (*The UniProt Consortium, 2017*) were initially used to search for the corresponding PDB entries to each of the UniProt IDs. For ECOLI, all UniProt IDs were checked to see how many PDB entries could be found for each protein, and then the protein list was sorted by amount of entries, showing that about 30% of the ECOLI proteins have at least one X-ray entry, and roughly a half of those have more than three entries. Drill down search by organism in the PDB was also used to supplement the protein list of *E. coli* outside the ECOLI proteome set. In fact, about 30% of *E. coli* PDB entries are of the UniProt IDs not found in the ECOLI set. These entries are used if they are of the structures of virtually identical sequences to any of the ECOLI set proteins. Thus, some of 980 xUMFs of ECOLI set are calculated from multiple UniProt IDs of almost identical sequences. For HUMAN, drill down search list was primarily used after UniProt ID sorting to find out which proteins are rich in the PDB entries. Roughly 23% (~4,700) of HUMAN reference set and reviewed proteins were found to have at least one X-ray PDB entry. This means that the number of targets amenable for DSA is fairly more than the presently performed ~1,000 if we carry out the analysis as extensively as has been done for ECOLI set.

### Archiving

Coordinate directories and data directories for proteins are separately made per UniProt ID under the corresponding organism top directory. Each coordinate directory contains

the raw pdb files assigned to the respective UniProt ID. The data directory of each protein holds the following basic file set: (1) list file, containing information of PDB entries (chain IDs, space group, lattice constants, resolution, etc), the polypeptide range (initial and final sequence positions) selected for DSA, and the order of entry/chain usage in the progress plot (*Anzai et al., 2018*). (2) Distance file, containing all $C_\alpha$ pair position numbers and the corresponding distances. (3) Score file, containing all $C_\alpha$ pair position numbers and the corresponding average distances, stdevs, scores, maximum and minimum distances. (4) Main plot scatter chart (and/or semi-log main plot scatter chart). (5) Progress plot table, and (6) progress plot scatter chart.

To date, over 200 species or group of species (~80 eukaryotic, ~110 bacterial, ~20 archaeal and ~15 viral) have been archived and the total number of proteins analyzed so far exceeds 5,500, although the degree of DSA completeness varies among the species. Completeness means how many multiple chains were included per entry. For example, our previous analysis of 300 human proteins used only one chain per entry because we selected the proteins having at least four entries. Although multiple chain usage from an entry do not always affect the UMFs substantially, there are some cases where many chain As from distinct entries are very similar to each other but chains A and B differs from each other. Thus, some of the xUMFs of human proteins would exhibit lower values when our chain usage completeness becomes higher.

The tables of protein lists associated with the present study are provided as Supplemental Files (Tables S1 and S2). The raw data files for *E. coli* and human proteins are stored in Mendeley Data.

An example of the list files (stored as list_****.xlsx in each protein's raw data folder) described above is provided as Table S3 in which the data of the cell O1, DSA123 (110–232), indicates that we have analyzed continuous 123 amino acid chains of the sequence number from 110 to 232. The numbers in the O column show the order of data inclusion and correspond to the x-axis of the progress plot (stored as ave-score_***.png).

## Data processing

DSA procedure was performed the same as previously described (*Anzai et al., 2018*). A Python script score-analyzer v22 has been modified from the previous version, but the way of calculation of $C_\alpha$ distances (Euclidian) remains unchanged, being compatible with commonly used protein crystallography tools. Among many minor changes, such as the output function of a semi-log main plot chart, for example, up to 1,000,000 $C_\alpha$–$C_\alpha$ pairs (over 1,400 amino acid length) can be processed now, covering almost all of the currently deposited and continuously modeled structures. The longest polypeptide analyzed so far is the human U5 small nuclear ribonucleoprotein 200 kDa helicase (1,397 residues, 65.4% of the full length is continuously modeled). In the previous study, the longest polypeptide analyzed was 919 amino acids (90.2% of the full length) of insulin-degrading enzyme.

The 7TM structures for CNR1, a G protein-coupled receptor, were obtained from our archive at gses.jp/7tmsp. Each of the chains contains 200 amino acids, defined

previously (*Okada, 2012*) for reproducible comparison of all receptors including non-rhodopsin-like GPCRs.

## Mapping of UMFs to the proteome set

The protein list of ECOLI and HUMAN reference (reviewed) sets obtained from UniProt were used as a template. Sequence lists (FASTA) of these reference sets were used for the online analysis by SCAMPI (*Peters et al., 2016*) to assign probable membrane proteins and the result was merged to the protein template list. The coverage of the present UMFs in the ECOLI set was evaluated first by trying to map all 1,178 *E. coli* xUMFs to any of the proteins on the template list, resulting in 980 non-redundant assignments. For cryo-EM data, 57 out of 58 eUMF were assigned to the proteins on the template list, including 31 redundant assignment with the xUMF. In many cases of these 31 proteins, eUMF was found to be substantially lower than the corresponding xUMF, regardless of the number of chains used for analysis. When we mapped all 1,043 xUMFs in the HUMAN reviewed reference set, no redundancy was found.

# RESULTS

## UMF of proteome set

Since many proteins are deposited to the PDB from a variety of *E. coli* strains, we chose a reference proteome set called ECOLI (strain K-12) in UniProt. According to UniProt reference proteome sets, the proteome size ranges from 4,232 (strain 333) to 6,593 proteins (strain NCTC13148). Thus, the number of proteins in the ECOLI reference set (4,391) is likely to be close to the minimum number necessary for constituting this prokaryotic organism.

As of July in 2019, the PDB contains over 8,500 X-ray entries that include *E. coli* components (mostly proteins). We calculated xUMFs for all *E. coli* strains, totaling 1,178 proteins, including redundancy in the case of some proteins. An example of this protein redundancy is β-lactamase, for which many UniProt IDs are found in the various strains of *E. coli*. Thus, 1,178 xUMF values were mapped to the reference proteome set ECOLI, resulting in 980 proteins or 22.3% (980/4,391) coverage of the whole reference proteome. On the other hand, when we re-calculated the coverage separately for soluble and membrane proteins assigned by SCAMPI, a significant difference was found; 26.9% (887/3,300) for soluble proteins and 8.5% (93/1,091) for membrane proteins. Thus, our present analysis provides xUMFs over a quarter of soluble proteins in *E. coli* K-12. The average ratio of the analyzed to the full length (including signal/propeptide sequences) for 980 proteins is 83.6% (only 6.7% of 980 proteins is limited to <50% of full length).

For comparison, we have updated the previous human protein analysis and collected 1,043 xUMFs. The SCAMPI analysis for the reviewed human reference set of 20,416 proteins assigned 6,293 as membrane proteins. Since 222 out of 1,043 xUMFs presumably are part of the 6,293 transmembrane (TM) proteins, the coverage is 3.6%; whereas, 821 out of 14,123 (=20,416–6,293) corresponds to 5.8% coverage for human soluble proteins.

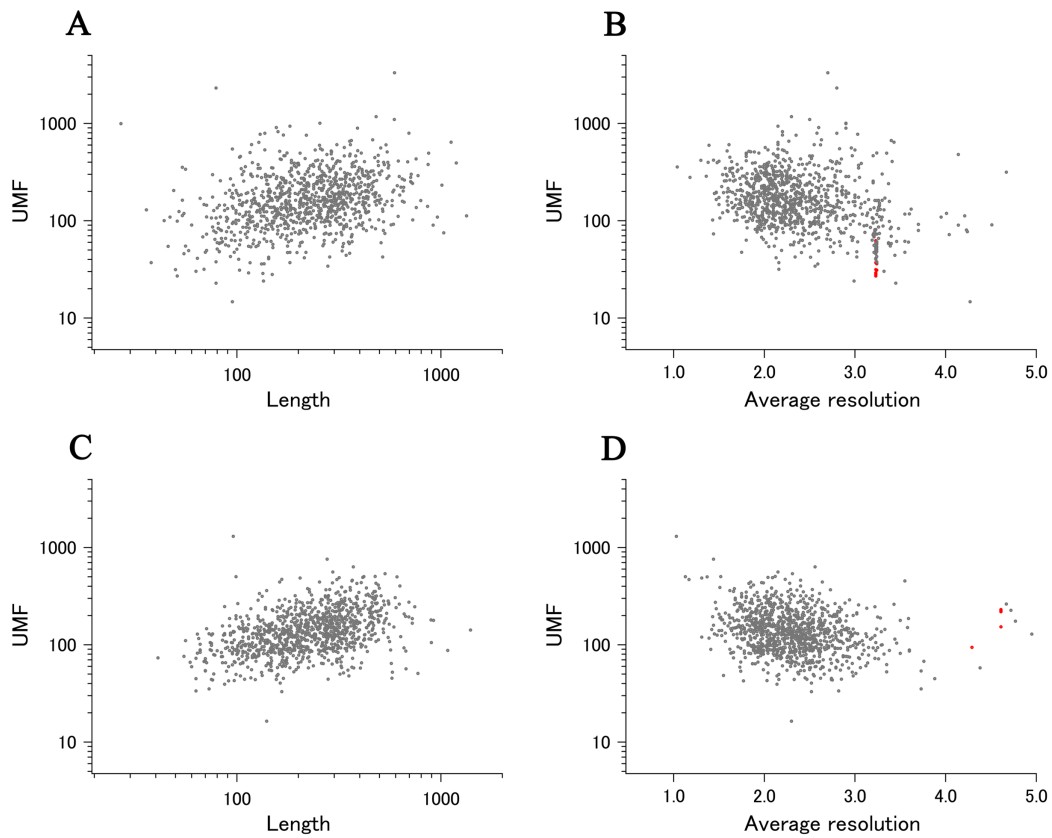

**Figure 2 The summary plot of xUMFs.** (A) Log–log plot of ECOLI set against analyzed chain length. (B) Semi-log plot of ECOLI proteins against average resolution. A total of 10 out of 52 ribosomal proteins are colored red, showing the UMF range roughly from 25 to 70. (C) Log–log plot of human proteins against analyzed chain length. (D) Semi-log plot of human proteins against average resolution (Å). Five COP9 proteins are colored red.

## Summary plot for *E. coli* and human proteins

When all 980 xUMFs of the ECOLI set proteins are plotted against the analyzed chain length in log–log scale, a fairly dispersed image can be obtained (Fig. 2A). This is a log version of the previously described "summary plot" (*Anzai et al., 2018*). A corresponding figure for 1,039 human proteins is also shown in Fig. 2C. These two images are similar despite all proteins being chosen randomly and only by giving selection priority based on the availability of the experimental coordinates by X-ray crystallography. An obvious difference between the ECOLI set and the human data is the amount of high xUMF proteins in the former and this presumably occurs because xUMFs of a substantial part of the ECOLI set proteins are calculated using less than three PDB entries; this means that more than one chain per entry had to be used. In fact, of the 38 proteins exhibiting xUMFs larger than 500, 34 proteins are analyzed using only one or two entries.

When the horizontal axis is changed to the average resolution, a cluster of proteins becomes clearer. In the ECOLI plot (Fig. 2B), ribosomal proteins, which includes thirty-two 50S and twenty 30S subunit proteins analyzed using mostly the same set of PDB entries, are shown as a thin cluster at the average resolution from 3.2 to 3.3 Å and the

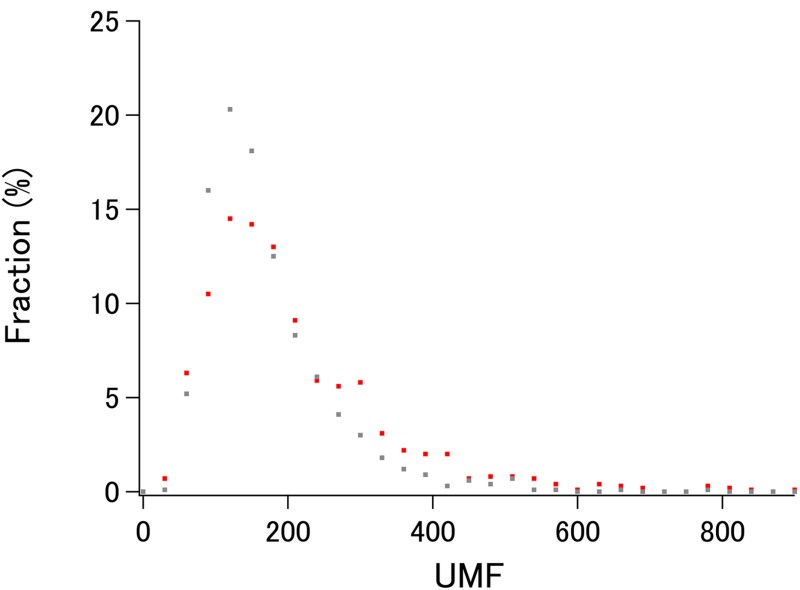

**Figure 3 The distribution plot of xUMFs.** Red: ECOLI set. Gray: human proteins.

xUMF range from 25 to 70. On the other hand, COP9 signalosome complex subunits dominate the data of average resolution lower than 4.3 Å in the human summary plot (Fig. 2D).

## Distribution plot

The summary plots of ECOLI and human proteins confirmed that the lower limit of xUMFs is ~15, as previously reported for human calmodulin. As long as we analyze X-ray structures, xUMFs in the range from 0 to 10 (over 10% average deviations for all intramolecular $C_\alpha$–$C_\alpha$ pair distances) are not likely to emerge. In fact, based on the over 5,500 proteins from various species analyzed so far, the lower limit of ~15 is still valid (unpublished result). Many proteins that are reluctant to crystallize would contain very flexible parts. If such flexible parts dominate the entire length of a polypeptide in a given protein, as can be seen in many NMR depositions (ensembles), the UMF should approach to 0. Thus, many intrinsically disordered proteins (*Dunker et al., 2001*; *Ahrens, Nunez-Castilla & Siltberg-Liberles, 2017*) can be understood to exhibit 0~10 UMFs that cannot be evaluated by X-ray crystallography.

To demonstrate the probable upper limit of xUMFs, distribution of xUMF values shown in the summary plot are counted from 0 to 900 with the slot of 30 (Fig. 3). Whereas some proteins (8 out of 980 *E. coli* proteins, and 1 out of 1,039 human proteins) exhibited the xUMFs larger than 900 and not shown in the figure, the plot demonstrates that most proteins would rarely exceed the xUMF over 500. This result means that the averaged uncertainty (standard deviation) of all $C_\alpha$–$C_\alpha$ distances in a folded polypeptide chain determined by X-ray crystallography inevitably exceeds 0.2% (0.1 Å per 50 Å).
## Cryo-EM structure analysis

Cryo-EM data deposition is remarkably increasing in the PDB (*Bai, McMullan & Scheres, 2015*; *Nogales, 2016*; *Raunser, 2017*). While the resolution of data by this method rarely reaches the precision that most X-ray structures can resolve, we have examined cryo-EM structures of ECOLI reference proteome set and calculated eUMFs for 56 proteins of which 31 have respective xUMFs. As a result, 1,005 UMFs (980 xUMFs plus 25 eUMFs) out of 4,391 ECORI set are obtained, corresponding to 22.9% coverage. Membrane proteins make up 11 of the 25 eUMFs, reflecting that cryo-EM is frequently applied to targets that are reluctant to crystallize.

When we add the eUMF data to the ECOLI reference set summary plot of xUMFs shown in Fig. 2A clear distinction becomes obvious (Fig. 4). Importantly, eUMF can be seen more frequently in the long polypeptides and in the low value ranges.

DSA analysis of cryo-EM structures also provided some examples that are distinct from that obtained from a larger set of X-ray entries. The semi-log main plot, which is another form of the previously reported main plot (*Anzai et al., 2018*) and is a 2D representation of highly variable parts in a protein, is especially favorable in demonstrating how cryo-EM and X-ray structures can explore the morphness of proteins. In Fig. 5A semi-log main plot is shown for the cryo-EM entry (PDB ID: 5Y4O) of low conductance mechanosensitive channel YnaI (*Yu et al., 2018*). This entry contains seven chains of identical length (223 residues, 65% of the full length). The unusually high eUMF of 79,821.6 on this plot is explained by a large part of the polypeptide being virtually identical among the seven chains and clearly separated from the lower score thin cluster representing the $C_\alpha$ pair distances in the TM part. Obviously, the eUMF in this case is far from the expected value for this protein. Seven other proteins are also outstanding in the unusual value of eUMFs (Fig. 4; Table S1).

YnaI is functionally related to the MscS mechanosensitive channel (*Böttcher et al., 2015*). Although the sequence similarity between them is limited, they both self-assemble to form a heptamer, in which each protomer assumes an elongated conformation. Single entry analysis of MscS as has been done for YnaI, but using X-ray entry (PDB ID: 2OAU) (*Steinbacher et al., 2007*), resulted in xUMF of 190.6, which is considerably lower than eUMF of YnaI and fairly in the range shown in the distribution plot (Fig. 3). The corresponding semi-log main plot demonstrated multidomain morphness (Fig. 5C), but not separated from each other in the average score range. Multi entry analysis (one chain usage per entry and six entry usage) lowered the xUMF to 119.0, resulting in a diffused pattern on the semi-log plot (Fig. 5B). For comparison, the pattern of semi-log plot obtained for 200 residue TM bundle of a G protein-coupled Cannabinoid receptor 1 (CNR1) is shown in Fig. 5D. As the analyzed part is only limited to compactly arranged seven TM helices, the maximum dimension is about a half of YnaI and MscS. This receptor, whose conformational change occurs mostly in one of the seven helices (*Hua et al., 2017*), is among many examples where any large domain morphness does not occur but the polypeptide contains very variable parts, resulting in a low xUMF (58.9 for CNR1).

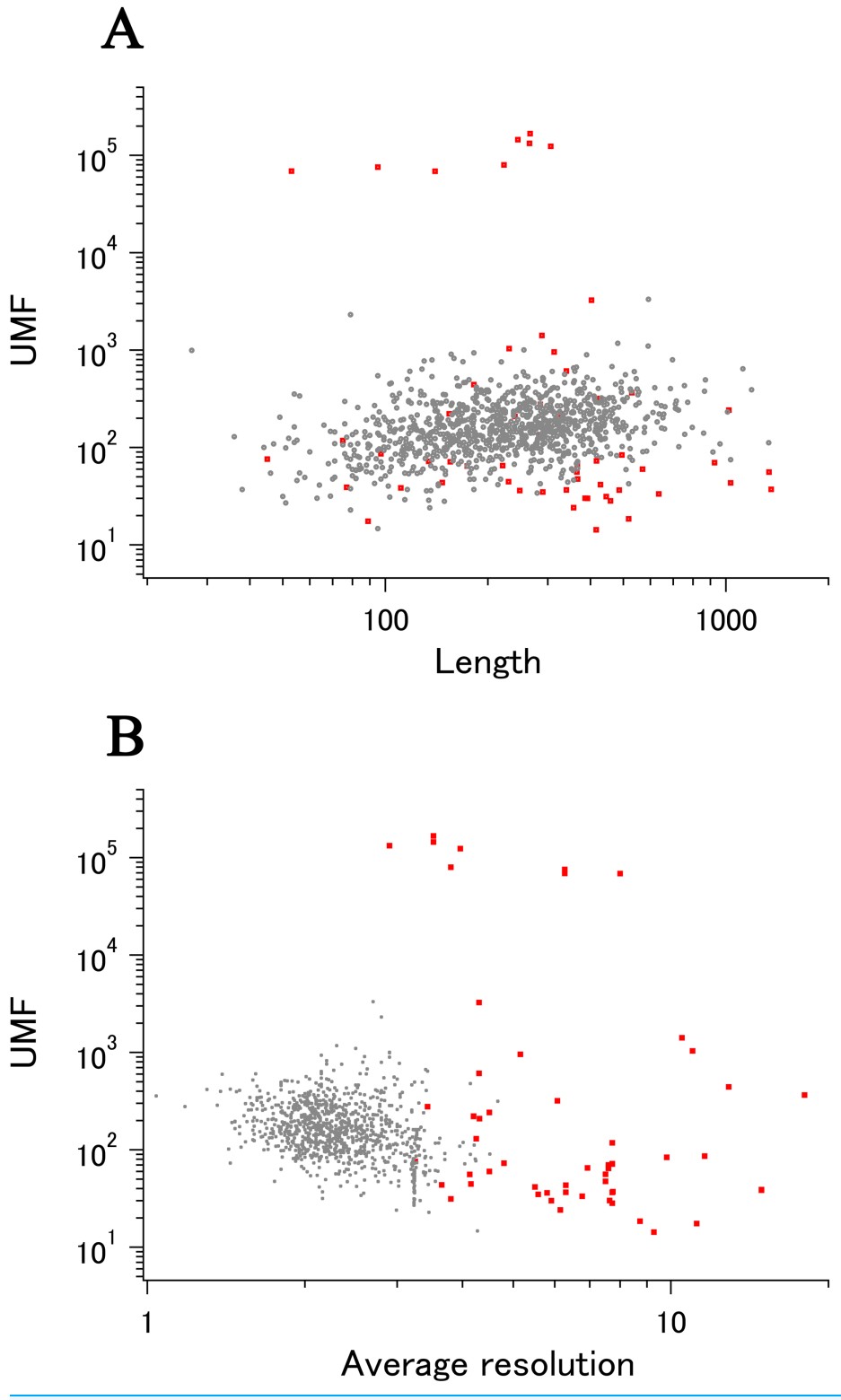

**Figure 4 The summary plot of xUMFs and eUMFs.** (A) Log–log plot of ECOLI set against analyzed chain length. (B) Log–log plot of ECOLI set against average resolution. eUMFs are shown as red squares.            

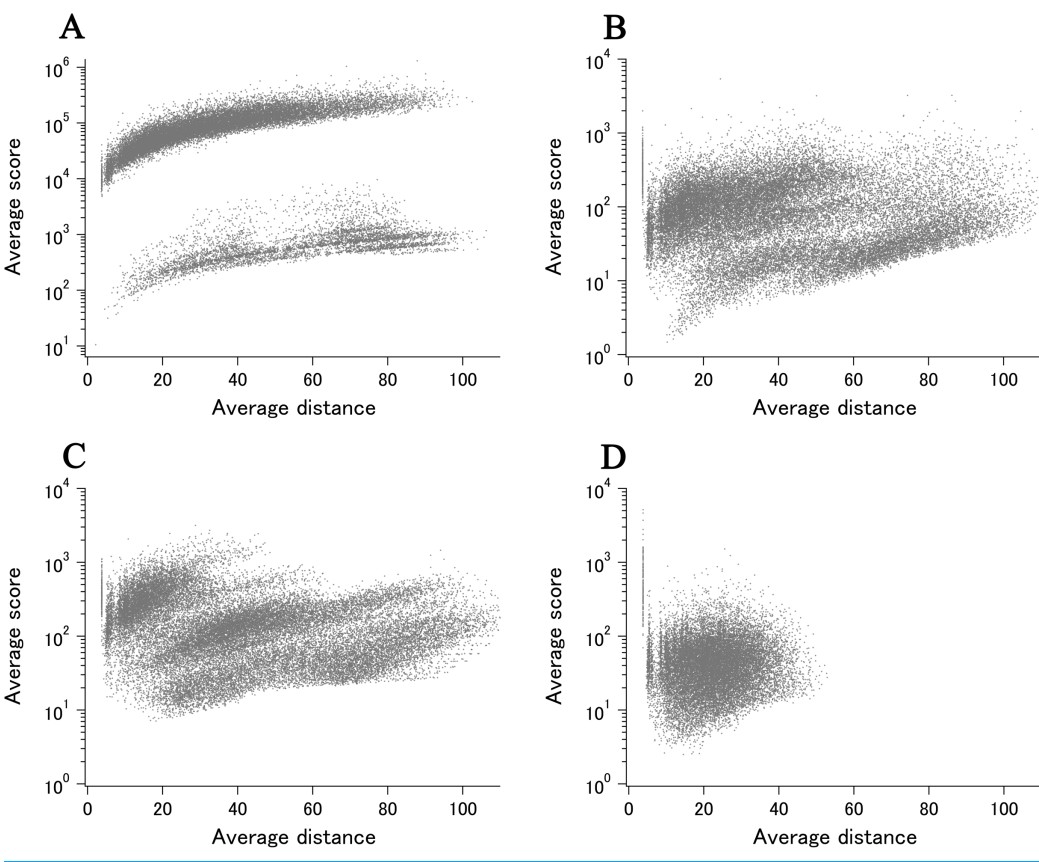

**Figure 5** **The semi-log main plot obtained from DSA.** (A) YnaI of ECOLI (cryo-EM). (B) MscS of ECOLI by multi entry analysis (X-ray). (C) MscS of ECOLI by single entry analysis (X-ray). (D) CNR1 of human (X-ray). The unit of average distance is Å.     

For other two proteins, ribosome hibernation promoting factor (HPF) and ribosome modulation factor (RMF) (*Beckert et al., 2018*), out of the eight proteins with unusually high eUMFs (Fig. 4), there were respective X-ray entries (but not enough chains for xUMF calculation). By merging just one X-ray entry with the corresponding cryo-EM entries for each of the two proteins, the resulting UMF (xeUMF) dramatically decreased to 90.4 for HPF and 66.0 for RMF, falling into range on distribution plot of ECOLI (Fig. 3, red). This result is promising to further utilization of increasing cryo-EM entries in combination with the respective X-ray entries for obtaining more UMF data for the proteins that were not amenable to DSA.

In the normal scale main plot of YnaI (Fig. 6A), $C_\alpha$ pairs in the morphable parts are difficult to assess when we include all data points. On the other hand, $C_\alpha$ pairs in the unmorphable parts exhibit almost linear distance dependency. This type of main plot did not typically arise from a set of X-ray structures, but could be seen for a polypeptide consisting of a larger multi-chain complex, such as human hist1h2ab (histone H2A.2) as demonstrated previously (*Anzai et al., 2018*). The updated main plot, adding eight entries (chains) to the previous data, of hist1h2ab is shown in Fig. 6C.

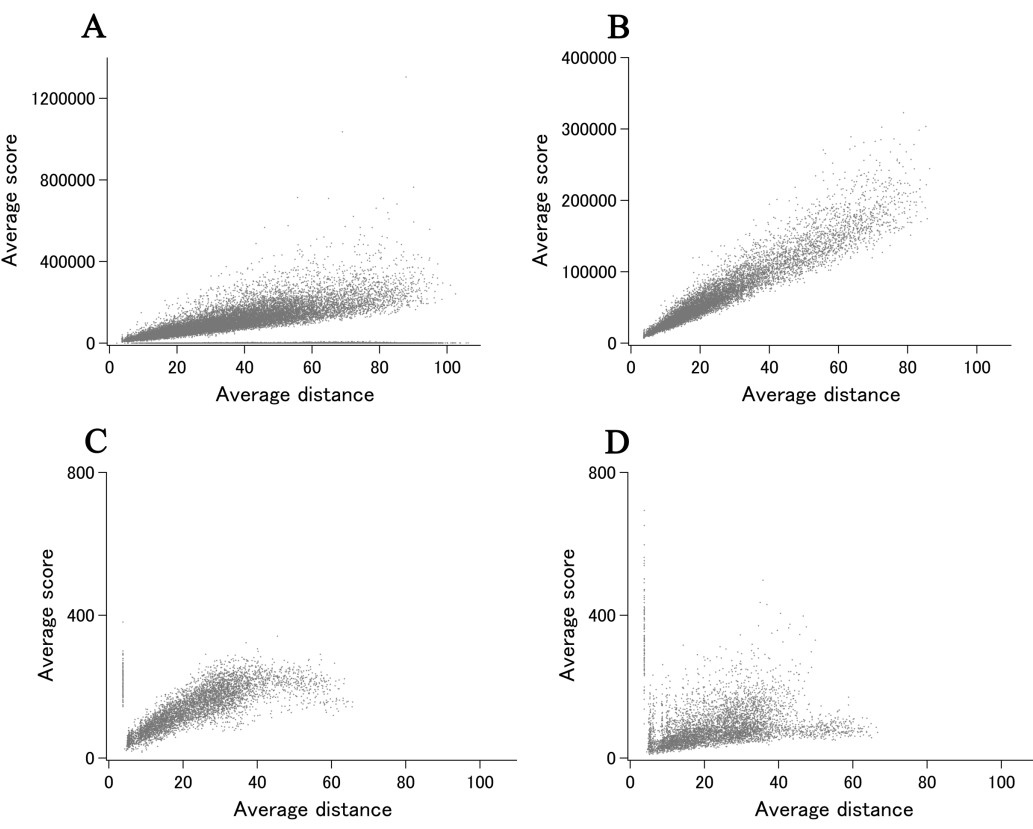

**Figure 6 The main plot obtained from DSA.** (A) YnaI of ECOLI (cryo-EM). (B) Prepilin peptidase-dependent pilin of ECOLI (cryo-EM). (C) Hist1h2ab of human (X-ray). (D) Hist1h2ab of human (cryo-EM).

For prepilin peptidase-dependent pilin (*Bardiaux et al., 2019*), which is another one of eight unusual eUMF value, the main plot (Fig. 6B) was found to be just like the pattern of hist1h2ab. These patterns are almost artificially reproducible by using values of model standard deviations for the average distances, and represent the examples of uniform $C_\alpha$ coordinate deviations at all positions in the polypeptide.

In Fig. 6D, the main plot of hist1h2ab by cryo-EM entries is shown. Although the used number of chains (22 chains, one chain per entries) are less than a half of X-ray entries (56 chains, one chain per entries), the cluster of $C_\alpha$ pairs by cryo-EM was found mostly in the lower average scores than that by X-ray data. Correspondingly, eUMF of 91.0 is lower than the xUMF of 143.5 which is substantially unchanged from the previous value with 48 chains usage.

## Application to recent entries

How newly added PDB entries exhibit different backbone structures from that of previously deposited ones for the same protein is readily evaluated by the present method. Here we demonstrate an example obtained recently in July of 2019.

SOS response-associated protein YedK (*Thompson et al., 2019*) is included in the ECOLI reference proteome set and has a calculated xUMF of 321.7 (shown as a red circle in Fig. 7A),

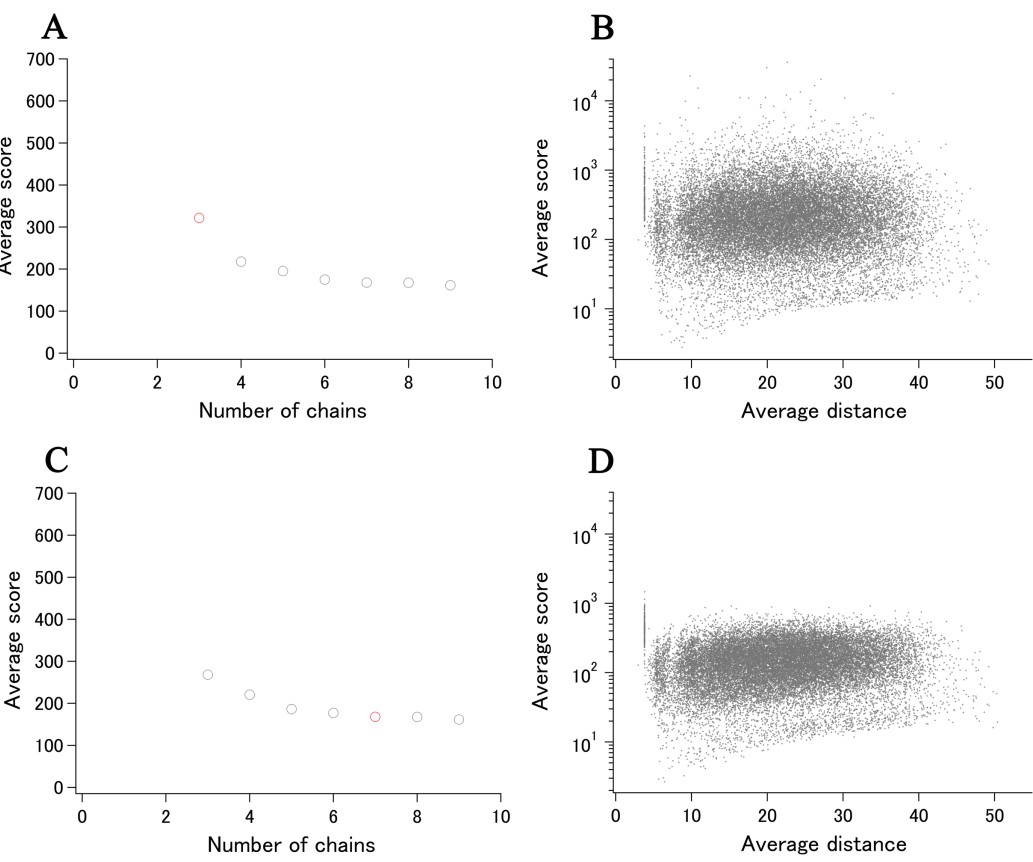

**Figure 7 Recent examples of DSA update.** (A) Progress plot of YedK (P76318/A0A2S5ZH06, the data point of only P76318 is shown in red). (B) Semi-log main plot of YedK (P76318). (C) Progress plot of YedK (P76318/A0A2S5ZH06, the third data point in (A) is moved to the 7th position). (D) Semi-log main plot of YedK (P76318/A0A2S5ZH06).

using only three chains from two entries out of three entries deposited for UniProt ID of P76318. The unused entry contains two regions of chain-break within the almost full-length model.

More recently, five entries were added to the PDB for the same protein but with different UniProt ID of A0A2S5ZH06 (100% identical to P76318). As the modeled length was the same as that of two entries for P76318, the new distance data can be added to the previous distance data, making the progress plot meaningful (shown as gray circles in Fig. 7A). The new data contributed to the data point 4–9, using four out of five entries and six chains from four entries (three chains from one entry 6 KBZ). The average score value at the data point 9 translates to xUMF of 161.4, which is about a half of the previous 321.7. We have been making clear separation of entry usage and chain usage for most of over 5,500 protein analysis, and the progress plot in this case is rearranged to six entry usage first followed by the redundant chain usage as shown in Fig. 7C where a redundantly used chain for P76318 is moved from data point 3 to 7. Although such a data switch in the progress plot did not cause any significant change in this case, data order switch

would be useful for other proteins to see the impact (discontinuous drop of the average score) of some specific entries/chains in a set of many entries.

Two semi-log plots, one for P76318 and the other for P76318/A0A2S5ZH06, demonstrate the effect of entry/chain addition (Figs. 7B and 7D). Most of the $C_\alpha$–$C_\alpha$ pairs that appeared in the average score of over 1,000, moved to much lower score region, whereas the highly variable pairs with low scores of less than 10 remain largely unchanged. This indicated that the first two entries (three chains) for YedK differed significantly from each other in the high morphness regions.

## DISCUSSION

In this study, proteins are selected for DSA when at least three chains (for *E. coli*) or three entries (for human) are available, with an exception for the case that the continuously modeled part is too short against the full length. Overall, the average usage per protein is 7.9 entries (10.1 chains) for the ECOLI reference set and 13.9 entries (15.4 chains) for the human proteins. The higher number in the human set is due to the data abundance in the PDB and to several heavily deposited proteins (such as carbonic anhydrase 2, for which we have used over 600 entries to calculate the xUMF). When we recalculate the numbers for the human set after removing the 12 proteins for which more than 100 entries/chains are used, the average usage drops to 10.9 entries (12.5 chains). For the ECOLI set, only maltose-binding periplasmic protein exceeds 100 entry/chain usage. These considerations indicate that the overall amount of data usage is not very different between the ECOLI set and the human protein set. On the other hand, a substantial portion of the ECOLI set xUMFs originates from the analysis of using only one or two entries per protein, taking multiple chains per entry. This limitation is presumably reflected in the xUMF distribution plot (Fig. 3) as discussed below. Our analysis also suggests that multiple chain usage frequently works for decreasing the UMF of a protein for which only one type of crystal lattice is available for multiple entries.

As for the summary plots (Fig. 2), xUMFs appear to exhibit only weak dependance either on the polypeptide length or on the X-ray data resolution (correlation coefficients of 0.19, −0.12, 0.30, −0.21 for panel A, B, C, D, respectively). We have previously reported correlation coefficients corresponding to Fig. 2D of the present study but for ~100 proteins × 3 sets separated according to the amount of entry usage. The previous value of −0.33 (an average of 3 sets) decreased substantially to −0.21 for the 1,039 proteins analyzed here. As a reference, similar analysis of over 5,500 proteins from various species (including *E. coli* and human) resulted in correlation coefficients of 0.28 and −0.22 vs. chain length and average resolution, respectively (unpublished results). These studies are consistent with a conclusion that xUMFs are slightly more dependent on the chain length than on the data resolution.

A clear difference between the ECOLI set and human proteins in the distribution plot (Fig. 3) is the smoothness of the pattern. In addition, the peak fraction is clearer in the human plot (gray), where over 20% of the 1,039 proteins are in the xUMF range of 90–120, corresponding to 1.1–0.83% average uncertainty of all $C_\alpha$–$C_\alpha$ distances. The higher fraction in the UMF range roughly from 250 to 400 in *E. coli* (red) could be due to the

proteins analyzed using more than three chains per entry. It should also be noted that the distribution plot of human was so irregular in the previous analysis of a limited number of proteins that it could not be considered (*Anzai et al., 2018*).

If we roughly classify the *E. coli* proteins as flexible, average, and rigid, based on the xUMF ranges 0–100, 101–200 and 201–, the ratio in each class is 22.9%, 43.0% and 34.1%, respectively (29.1%, 49.7% and 21.2% for human proteins). Differences in the xUMF values among the proteins analyzed by DSA reflect the presence/absence of distinct conformational states and/or the degree of backbone fluctuation in each protein.

Our results indicate that cryo-EM is effective in detecting conformational variability of large molecular weight proteins (Fig. 4). While the clear differences in the main plot pattern between cryo-EM and X-ray (Figs. 5 and 6) could arise from various factors, our results suggest that cryo-EM is potentially effective in uncovering the full morphness range of a polypeptide with fewer structure determinations (PDB entries).

## CONCLUSIONS

The growth in the structural data accumulation has been outstanding for human proteins and the trend will undoubtedly continue in the future. Uncovering the morphness repertoire for any eukaryotic organism is very challenging due to the presence of splice variants and intrinsically disordered proteins. The total number of PDB entries for *E. coli* proteins is the second largest (after human proteins) and is exceptionally large among prokaryotic organisms. The present study shows that about a quarter of *E. coli*'s proteome set could have already been the subject of detailed morphness analysis. More coverage would be possible if we further include very similar orthologues from other bacteria such as *salmonella* species. UMF calculation by cryo-EM data only (eUMF) was found to be unreliable in some cases but the data would undoubtedly be useful especially in combination with the corresponding X-ray data. It should be noted that we have not applied any resolution cut criteria for cryo-EM data analysis. While NMR data usage has not been given priority in our studies so far, it will have to be done in the future.

All DSA results can improve on a weekly basis in accordance with updates of the PDB contents. Importantly, the results presented here, when the sharing environment is established, can conform to FAIR data principles (*Wilkinson et al., 2016*) since no assumptions are involved in the analysis. Anyone can contribute to improve the quality of the data set by, for example, rationally changing the selection of the entries/chains based on their expertize of specific proteins.

### Funding

The authors received no funding for this work.

### Competing Interests

The authors declare that they have no competing interests.

## Author Contributions

- Keisuke Izumi performed the experiments, analyzed the data, authored or reviewed drafts of the paper, and approved the final draft.
- Eitaro Saho performed the experiments, analyzed the data, authored or reviewed drafts of the paper, and approved the final draft.
- Ayuka Kutomi performed the experiments, analyzed the data, authored or reviewed drafts of the paper, and approved the final draft.
- Fumiaki Tomoike performed the experiments, analyzed the data, prepared figures and/or tables, authored or reviewed drafts of the paper, and approved the final draft.
- Tetsuji Okada conceived and designed the experiments, performed the experiments, analyzed the data, prepared figures and/or tables, authored or reviewed drafts of the paper, and approved the final draft.

## Data Availability

Data is available at GSES: http://www.gses.jp/7tmsp/index.php?Top.

Data is also available at Mendeley:

*E. coli* (A–M, by gene name): Okada, Tetsuji (2019), "*E. coli* (A–M, by gene name)", Mendeley Data, v2 DOI 10.17632/92p6r2kbcw.2.

*E. coli* (N–Z, by gene name): Okada, Tetsuji (2019), "*E. coli* (N–Z, by gene name)", Mendeley Data, v2 DOI 10.17632/nh5sg68bnh.2.

Human (A–M, by gene name): Okada, Tetsuji (2019), "human (A–M, by gene name)", Mendeley Data, v1 DOI 10.17632/6cs36dvw2r.1.

Human (N–Z, by gene name): Okada, Tetsuji (2019), "human (N–Z, by gene name)", Mendeley Data, v2 DOI 10.17632/2y69rv4yb4.2.

## Supplemental Information

Supplemental information for this article can be found online at http://dx.doi.org/10.7717/peerj.8606#supplemental-information.

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
