# Peer review of "Repertoire of morphable proteins in an organism"

_PeerJ, doi:10.7717/peerj.8606_

## Round 0.1 · original submission · Major Revisions

Please take into consideration all the concerns from the reviewers and prepare a detailed rebuttal letter addressing all their questions.

Reviewer 1 ·

Basic reporting

The abstract is not informative about the paper's contribution which is stated at a general level. I´ll try to give more details about this in the Comments to the authors section.

Experimental design

The structure of the experiments needs to be guided by specific objectives or specific questions the authors would like to answer.

Validity of the findings

The current results and analysis are not enough to see the impact it can have on structural biology research. A clearer explanation of how this work differs from the one of Anzai et al. 2018, should be given.

I will suggest to the authors to provide a rationale for working with the protein’s chains separately. I believe this could work for a certain type of structures but for others also C-alpha distances between different chains will be required to characterize the structure. I will expect the authors to contribute to the explanation in what cases their approach will have different outcomes by using a single chain versus all chains.

Please also explain why the approach is not just another way to characterize secondary structures, for instance, you can compute the UMF for alpha helices and beta sheets and see how much they contribute to the total UMF. Like this, there are many aspects that the authors need to analyze to see a clearer picture of their contribution.

Additional comments

Some additional observations are:

ABSTRACT

By reading the abstract it is hard to understand what the actual contribution of the work is. For instance, it is not clear whether or not the main tool of analysis, the UMF, is proposed in this work.

Furthermore, at the end of the abstract it is stated ¨ The data and the method
presented here also conform to FAIR data principles,¨ however, the abstract does not present said method.

The problem statement is vague ¨ Efforts in the field of structural biology have contributed to uncovering the shape and the variability of each component¨

It will help to clearly state what is the open question this research address right from the abstract

Also, statements like ¨The distribution of UMFs obtained for an organism is expected to represent how rigid and flexible components are balanced. ¨

In the paper, you do not show how your approach explains the flexibility or rigidity of a protein.

INTRODUCTION

In the last paragraph of this section the actual contribution of the paper is starting to become clearer.

ARCHIVING

In the supplementary files, there are only 2 .xls all other files mentioned should be also included.

Mapping of UMFs to the proteome set

What is SCAMPI, please provide a reference for it the first time you introduce it.

Reviewer 2 ·

Basic reporting

There are typos and unclear sentences that need to be corrected. See general comments for the author.

Experimental design

The method that the authors are developing is solid and reliable.

Validity of the findings

See general comments for the author.

Additional comments

The authors have applied their own developed methods to the EMS data and compared the X-ray data. Generally, the findings in the manuscript is interesting but, I have already mentioned, there are typos and unclear sentences that need to be corrected. The reviewer suggests that the manuscript should be revised before publication.

1.
The abbreviations, xUMF and eUMF, are used on L.67 but they are defined at L.130-131. They are important key words in the manuscript and the definition of these special words before L.67. The word, xeUMF, is found on L262 and this word is not defined in the manuscript.

2.
L36-45, L126-131
The advantage of the “UME” score comparing to DSA score is not clear from the text.

3.
The captions of Fig2 (B) and (C) are probably exchanged, because Fig2 (B) is a semi-log plot and Fig2 (C) is a log-log plot. In Fig2 (B) (I think this figure is “semi-log plot of ECOLI set”) the cluster of ribosomal (mentioned on L181-183) should be clearly indicated ( for example colored). COP9 in Fig 2D (L184-135) is also indicated.

4
The scale of longitudinal axis: “10, 100, 1000” are used in Fig. 2 and 4, and “10^1, 10^2, 10^3, …” are used in Fig. 5. A same scale representation is better.

5
To make one figure of ECOLI set (Fig. 3A) and human protein (Fig. 3B) is helpful to see the difference. For the scale of longitudinal axis. % is better (In L.237-242, the authors discuss using %).

6
The caption of Fig.4 tells “eUMFs are shown as black squares” but it is difficult to identify these black squares. Make larger or change the color.

7
In L241-242 authors wrote “Four other proteins are also outstanding in the unusual value of eUMFs”. What are other 3 (or 7, see below) proteins?
I see 8 deviated values in Fig 4A and B (near 105 in the longitudinal axis). In L259 authors wrote “… out of the eight proteins with unusually high eUMFs …”. Is the correct number 4 or 8? And which points correspond to which proteins?

9
Minor:
In caption of Fig 3: plotof -> plot of
In caption of Fig 5: obtainedfrom -> obtained from
In caption of Fig 6: plotobtained -> plot obtained

Reviewer 3 ·

Basic reporting

The work entitled “Repertoire of morphable proteins in an organism” by Izumi and collaborators describes the use of an index (DSA) to describe structural variations in the PDB reported for the same protein that belong to the same organism, in this case Escherichia coli and Homo sapiens.

The work is in generally well written, yet there are several important aspects that need to be clarified in order to clarify the relevance of this work. These concerns will be described in the following sentences.

- The authors refer to a UMF in the abstract to refer to “experimentally observable structural variation”; in the text and methods, there is no single reference as to how to calculate such UMF, even though, the authors used this term to present their results in several figures. On the other hand, the authors described the DSA (distance scoring analysis) and described the general formula and advantage of its use. How UMF is related to DSA is not clear from their description. One can assume that figure 1 describes what UMF is, but it is not clear from the figure legend “Scheme of score calculation” or the description of the figure. I’ll assume for the sake of arguments that the all intramolecular Ca-Ca pair distance variations (score = media (distance(Ca-Ca))/stdev) is the UMF.

- Figure 6 and its description do not contribute much to understand protein movement or morphness. For instance, linear dependency could means that proteins did not suffer any rotation or large motions of some of their parts, like it is observed in multidomain proteins. In single-domain proteins, it is less likely that such movements take place, hence the no-linearity may emerge from low resolution at the N and C terminus end of the protein, that may vary from one crystal structure to another. This level of analysis may improve the level of relevance of their score that can capture these properties.

Experimental design

- The methods do not provide sufficient information to understand or reproduce how the authors calculate the UMF. For instance, the procedure to calculate Ca-Ca distance is not described. The reference from Anzai referred to another work of this group by Asano and colleagues, where they state that "The overall pairwise root-mean-squared deviation and correlation coefficient were obtained by Discovery Studio Visualizer 3.1 (Accelrys Inc.) and pca-excel 1.0 (ss-nakano Inc.), respectively". Discover Studio Visualizer is commercial software that although is freely available, does not specify the type of distance that calculates. There are several distances that can be obtained from an euclician space (see for instance: https://en.wikipedia.org/wiki/Distance). The most common one is referred to as euclidian distance, which is a 2-norm distance. In this case, the authors have not specified what type of distance they used for their studies. This is an important aspect since from the description of their work, their interest is to measure the distribution of variations. For that goal, there are different approaches that may be appropriate, such as divergences. If the authors are not using that distance, it would be relevant to justify why to use the Euclidian distance. This is relevant for instance, in terms of computational efficiency, there are important differences on how fast these distances can be computed. If the database growth surpasses the current capacity of their method, this may become an issue later on and perhaps the authors would need to change their distance. It is expected that changing the distance may change their results and consequently their conclusions. Thus, it is fundamental for the authors to make a better description of their procedure to compute distance and make an appropriate justification of their procedure.

- The previous point is relevant for the results presented in figure 3. The plots show a Poisson-like distribution, a form of normal distribution for small sets (see for instance https://en.wikipedia.org/wiki/Poisson_distribution). This distribution is possible to be induced by the way the authors processed their data. That is, the authors used the media of the Ca-Ca distances and divided this by the stdev; both media and stdev, are parameters of normal distributions. As the authors claim in favor of the DSA “the amount of numeral usage is far more than the pair-wise based analysis”; with more numbers to analyze, it is feasible to analyze the distribution of their values (Ca-Ca distance values) to test whether these follow the observed distribution in Figure 3 or not. If these distances do not follow the distribution of Figure 3, then their treatment of this data is likely inducing a behavior that is not present in the original data. This in turn may have an effect on the author’s results and conclusions. Hence, it is important to show this analysis of distances to test for the lack of induction of their method.

- Furthermore, the Ca-Ca distance distribution would allow for better interpretation of their data. For instance, in the section “Distribution plot”, the authors described how the UMF values vary from a minimum of 15 to a maximum; however, there is no interpretation of these values to the protein itself. For instance, contiguous Ca-Ca distances are usually between 6-12 A apart; hence the limit for UMF=15 is probably the consequence of this range. The authors also mentioned the value close to 0 for unstructured proteins. Yet, considering that UMF=DSA, we can conclude that for unstructured proteins, the DSA would have large average distances and potentially low stdev, hence, that value would tend to be large not close to cero. But looking into their Ca-Ca distance may solve this apparent discrepancy. The interpretation of their results may focus on the protein structure and not so much on the UMF value itself. For instance, since the distance between alpha carbons cannot be extended too much, it seems that the upper limit of the DSA depends on how small the stdev is. Understanding why different structures of the same protein maintain the same orientation of each pair of alpha carbons may depend on the crystallographic conditions used. If the different structures were crystallized with the same procedure and obtained very similar resolutions, this may explain this small variation. The authors do not discuss this.

Validity of the findings

- The term morphable and its score are not clear what they try to capture from the protein structure. Considering the formula and their description, it seems that the authors’ interest is to capture the structural divergence between different experimental conditions used to solve the same protein. If that is the case, I would expect a clear relationship between their UMF score and a similar one derived from the crystallographic B (thermal) factors. However, the authors did not present any data to show this possible relationship. In any case, it is important for authors to show that their values reflect some aspect of the experiments to solve the structure, the nature of the protein or both. The current work lacks any analysis in this regard. For instance, the authors could extract information that is part of the PDB entries, such as CAVEAT, COMPND, SOURCE, AUTHOR among others, to infer how different the conditions used to solve two or more structures were. If there is a correlation between these experimental differences and the UMF, then this last score would be a valuable resource to resume lots of experimental information. The same applies for structural changes of the protein that reflect the structural flexibility of the protein as noted in the previous point.

Additional comments

Minor concerns
- Figures are presented in the text not in the order in which they are numbered; for instance, figure 7 is refer first than figure 3, which in turn is referred first than figure 2.

- line 102 “are now can be processed” may be clearer to say “can be processed now”.

- lines 117 to 120 in Methods, refer to results, not to methodology. This has to be changed.

- line 129 states “UMF converges as the number of used chains increases”; it is not clear to what converges to the UMF.

- line 141 “and 22.3%” would be better as “or 22.3%”.

-line 151 states that “Since 222 out of 1043 xUMFs are of the possible 6293 membrane proteins,”; it may be clearer to state “Since 222 out of 1043 xUMFs presumably are part of the 6293 transmembrane proteins,”.

- Figure 1; in the text the authors claim to have used the average distance between all pairs of alpha carbons, the image only shows the score for 1 pair, so it does not illustrate what the score is.

- Figure 2. The legends of the figure do not correspond with the plots.

- Figure 7 A and C, the symbols are too small.

- Figure 7. It is not clear what the authors mean by "This value appeared to be possible to converge to a lower one, whereas even the progress plot (Fig. 7A, C) could not be made with just one data point (number of chains = 3). The unused entry contains two regions of chain-break within the almost full-length model.".

---

## Round 0.2 · Major Revisions

Please review the comments on the revised version, and provide a detailed rebuttal letter addressing all questions.

Reviewer 1 ·

Basic reporting

'no comment'

Experimental design

'no comment'

Validity of the findings

'no comment'

Additional comments

The paper has been improved, however, additional improvements are required so that the reader can fully understand your contribution and other researchers can reproduce your results. The main concern is that I still find it difficult to reproduce what was done here, and I will suggest the authors to address all their effort in making every step as clear, detailed and self-contained as possible. Some examples of things that need to be improved are listed below.
1. You need to provide a clear, mathematical definition for UMF, with an example for it and the reference that proposed it for the first time, which I understand is due to Okada, 2018.
2. According to your definition what is considered flexible and what is rigid? Does it mean that if you have 20 experimental structures for the same protein and all of them represent the same conformation then this protein is rigid? Please give a physical explanation of the rigidity (flexibility) that is measured by the xUMF?
3. How do you support the following statement?
“ demonstrating how rigid and flexible components are balanced. ”
Can you provide a more detailed explanation about the composition of rigid and flexible components for the E. Coli proteome? How many proteins are rigid, how many proteins are flexible, what is the threshold in xUMF that separates one from the other?

4. It is not clear if only continuously modeled proteins are included, and if by this you mean proteins whose chains have no missing amino acid between the N-terminus and the C-terminus? The readers do not need to read Anzai´s work to discover this.

5. It seems that there is a contradiction with this sentence in the abstract ¨By collecting a new index “UnMorphness Factor” (UMF), proposed in this study and determined from DSA for each of the proteins,¨ and the following one in the Introduction ¨We previously proposed that “UnMorphness Factor” (UMF), (Okada, 2018) which is defined as a converged average score for a protein, is an alternative to conventional RMSD-based comparison.¨
The abstract says that you proposed UMF in this study while the Introduction suggests that it was proposed in Okada, 2018. Please clarify, and if needed correct.

Minor details,

English still needs to be carefully revised. For instance, in the last paragraph of the Introduction you need to change ¨ The present work is intended to assign an UMF¨ by ¨The present work is intended to assign a UMF¨
In the first paragraph of the Methods section
¨showing that about 30% of the ECOLI proteins have at least one X-ray entries¨ Vs. ¨showing that about 30% of the ECOLI proteins have at least one X-ray entrY¨

---

## Round 0.3 · accepted · Accept

The manuscript has improved over the review rounds and it is now accepted at PeerJ.

Reviewer 1 ·

Basic reporting

See ¨General Comments for the authors¨

Experimental design

See ¨General Comments for the authors¨

Validity of the findings

See ¨General Comments for the authors¨

Additional comments

The authors have answered and properly addressed all of my comments.